# Interelectrode Distance Analysis in the Water Defluoridation by Electrocoagulation Reactor

**Jesús Fernando Martínez-Villafañe** [1], **Juan Carlos Ortiz-Cuellar** [1], **Jesús Salvador Galindo-Valdés** [1], **Francisco Cepeda-Rodríguez** [1], **Josué Gómez-Casas** [1], **Nelly Abigaíl Rodríguez-Rosales** [2], **Oziel Gómez-Casas** [1] **and Carlos Rodrigo Muñiz-Valdez** [1,*]

1 Faculty of Engineering, Autonomous University of Coahuila, Blvd. Fundadores km 13 University City, Arteaga 25350, Mexico

2 Departamento de Metal Mecánica, Tecnologico Nacional de Mexico/I.T. Saltillo, Saltillo 25280, Mexico

* Correspondence: rodrigo.muniz@uadec.edu.mx

**Abstract:** In this research, the effect of the interelectrode distance (d) in the electrocoagulation (EC) reactor was studied. The experiments were carried out with varying *d* in values of 3, 5, and 9 mm during the treatment of water contaminated with fluoride ($F^-$). The response variables analyzed were the treatment time necessary to reduce the residual concentration of $F^-$ to 1.5 mg $L^{-1}$, the number of aluminum hydroxides formed, the potential drop in the reactor terminals, and the electric power consumption of the reactor. The software FLUENT version 6.3 was employed to simulate the liquid velocity profiles achieved in the reactor chamber. The results obtained show that the liquid velocity increases in the interelectrode spaces to 0.48, 0.65, and 0.86 m $s^{-1}$ for interelectrode distances of 9, 5, and 3 mm, respectively, which favors not only the formation of flocs but also the elimination of fluoride. With a shorter interelectrode distance, the EC reactor not only consumes less electrical energy but also fewer electrodes, and the dispersion of generated flocs in the reactor chamber is major, which is more important than the quantity of flocs generated in it.

**Keywords:** electrocoagulation; interelectrode distance; liquid velocity profiles; defluoridation

## 1. Introduction

Electrocoagulation (EC) is a technique that has been successfully used to treat wastewater [1–8], groundwater [9–15], and drinking water [16,17]. The technique uses a reactor fed with electric current to produce hydroxides in situ (generally of iron or aluminum) by the anodic oxidation of these metals in an aqueous medium [18]. The species formed in the reactor adsorb the contaminants found in the water and are removed by techniques such as sedimentation and filtration [18,19]. Disadvantages noted for EC reactor operation are the consumption of electrodes [19,20] and electrical energy [19–21]. The distance between electrodes is the parameter with the greatest influence on energy consumption [22]. Therefore, this study aims to determine the effect of different distances between electrodes: the treatment time required to reduce the residual concentration of $F^-$ to 1.5 mg $L^{-1}$ (the maximum level allowed by the World Health Organization in water intended for human consumption) [23], the energy and electrode consumption during the treatment time, and the liquid velocities profiles developed in the EC reactor.

## 2. Reactions in the Water Defluoridation Process

In the EC process, water defluorination is carried out by the following reactions [2,14,16]:
On the anode:

$$Al \rightarrow Al^{3+} + 3e^- \tag{1}$$

On the cathode:

$$3H_2O + 3e^- \rightarrow 1.5H_{2(g)} + 3OH^- \tag{2}$$

Aluminum ions react with the hydroxyl ions produced at the cathode to form aluminum hydroxide according to the following reaction [14,16]:

$$Al^{3+} + 3OH^- \rightarrow Al(OH)_{3(s)} \tag{3}$$

The aluminum hydroxides produced are colloidal solids that remove the $F^-$ and precipitate during the process. The reaction that describes the mechanism of $F^-$ removal by flocs Al (OH)$_3$ is an ion exchange reaction [14,16]:

$$3Al(OH)_{3\ (s)} + xF^- \rightarrow 3Al(OH)_{3-x}F_{x\ (s)} + xOH^- \tag{4}$$

### 3. Aluminum Hydroxides Generated (*w*)

The quantity of the aluminum hydroxides generated depends directly on the consumption of the electrodes, which are formed from the dissolution of the electrodes, as shown in reactions (1)–(3). The following equation is used to calculate the consumption of electrodes (*w*) in each water treatment process [1,24]:

$$w = \frac{ItM}{nF} \tag{5}$$

where (*I*) is the applied current, (*t*) is the treatment time, (*n*) is the number of electrons transferred, (*F*) is the Faraday constant, and (*M*) is the atomic mass of aluminum. According to this expression, the consumption of the electrodes is directly proportional to the treatment time and the supplied electric current.

### 4. Potential Drop on the Terminals (*U*)

The potential drop on the terminals, or the voltage drop across the reactor, is calculated using the following formula, disregarding the resistivity of the electrodes [24]:

$$U = E_A + \eta_A - E_C + \eta_C + d \cdot i \cdot \rho \tag{6}$$

where (*U*) is the voltage at the reactor terminals. This is equivalent to the sum of the individual potential differences involved in the process, as well as the ohmic drop of the aqueous medium ($d \cdot i \cdot \rho$), where (*d*) is the interelectrode distance, ($\rho$) the electrolyte resistivity and (*i*) is the current density, ($\eta_A$) and ($\eta_C$) are the overpotentials of anode and cathode, and ($E_A$) and ($E_C$) the thermodynamic potentials. However, this value can be obtained experimentally by measuring the voltage at the terminal electrodes of the reactor.

### 5. Electric Power Consumption *I*

The electrical energy consumed during the EC process is obtained from the potential drop in the reactor (*U*), the applied current (*I*), and the treatment time (*t*), as indicated in the following equation [24]:

$$E = I\ U\ t \tag{7}$$

### 6. Materials and Methods

The fluoride ($F^-$) water was prepared from a stock solution of sodium fluoride (NaF) and deionized water with 2.0 mM NaCl to simulate natural water with ionic strength. The pH value (7.0) was measured with a Hach pH meter.

The $F^-$ initial concentration ($C_{F0} = 5$ mg L$^{-1}$) and the samples obtained during the experimentation were measured using the potentiometric technique, using Orion 9609 ion selective electrodes. To avoid the interference of other ions during the analysis of fluoride, a TISAB II ionic strength-adjusting solution was used.

Three batch reactors with a volume of 1 L were used for the experiments, and electrode racks with vertically oriented rectangular aluminum plates with distances between them of 9, 5, and 3 mm were used. The electrode's dimensions were $10 \times 14 \times 0.635$ cm in height, width, and thickness, respectively, and connected in bipolar mode.

The current density was fixed at 3.0 mAcm$^{-2}$, employing a variable current source of 0–5 A.

To generate the liquid movement, porous membrane diffusers located at the base of the reactor insufflated bubbles of approximately 2 mm in diameter at a velocity of 0.3 m s$^{-1}$ by a flow air pump (Azoo 9500) of 4.0 L min$^{-1}$ discharge. The distribution of air bubbles was uniform throughout the reactor chamber, as can be seen in Figure 1.

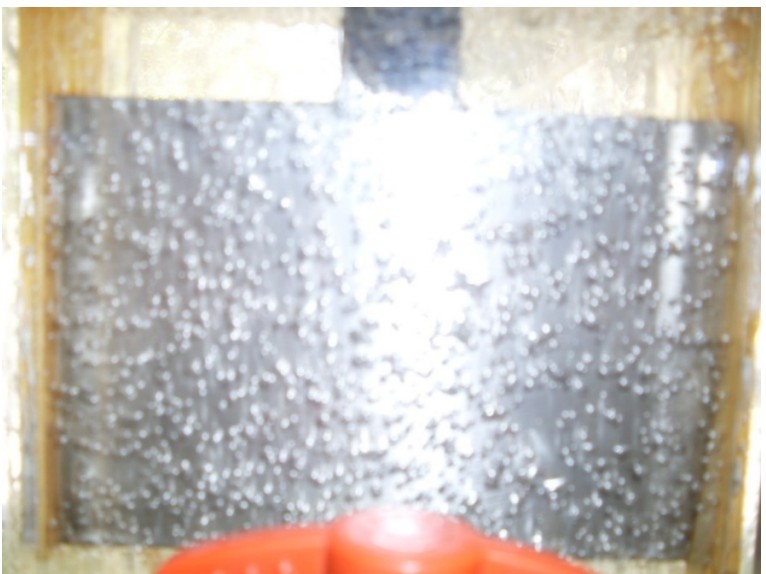

**Figure 1.** Distribution of air bubbles throughout the reactor chamber.

The precipitates obtained in the removal of F$^{-}$ by the EC process were separated from the treated water by cellulose filter paper with a pore size of 2.5 μm, subsequently dried for 2 h at 100 °C in a Furnace Thermolyne 1500 muffle, and were weighed using an Ohaus electronic analytical balance.

## 7. Results and Discussion

Figure 2 shows the effect of the interelectrode distance on the treatment time necessary to achieve $C_F$ = 1.5 mg L$^{-1}$.

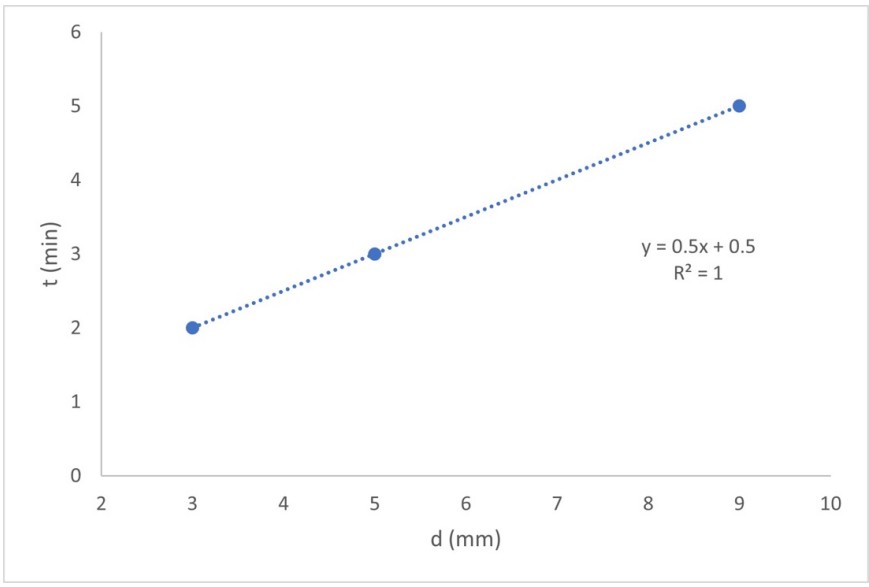

**Figure 2.** Effect of the interelectrode distance on the treatment time.

As shown in Figure 2, the treatment time is precisely adjusted to the ratio y = 0.5x + 0.5. The shortest treatment time was that corresponding to $d = 3$, followed by $d = 5$ and finally d = 9, with 2, 3, and 5 min, respectively. This is because reactions (3) and (4) are faster, due to the liquid velocity increases, and greater mass transfer (MT) is achieved in the reactor chamber. Therefore, it can be established that, with a smaller $d$, it will have a shorter treatment time necessary for the formation of aluminum hydroxides, which adsorb the $F^-$ present in the water (according to reactions (3) and (4)). On the other hand, during the experiments with the shorter $d$ values, not only the turbulence in the interelectrode spaces but also in the reactor chamber was greater, producing the dispersion of the $Al(OH_3)$ flocs in the reactor chamber and favoring the kinetics of these reactions.

Figure 3 shows the amount of aluminum hydroxides formed ($w$), obtained at the time of the treatment required to achieve a concentration of 1.5 mg $L^{-1}$ of $F^-$ in the treated water as a function of the interelectrode distance $d$.

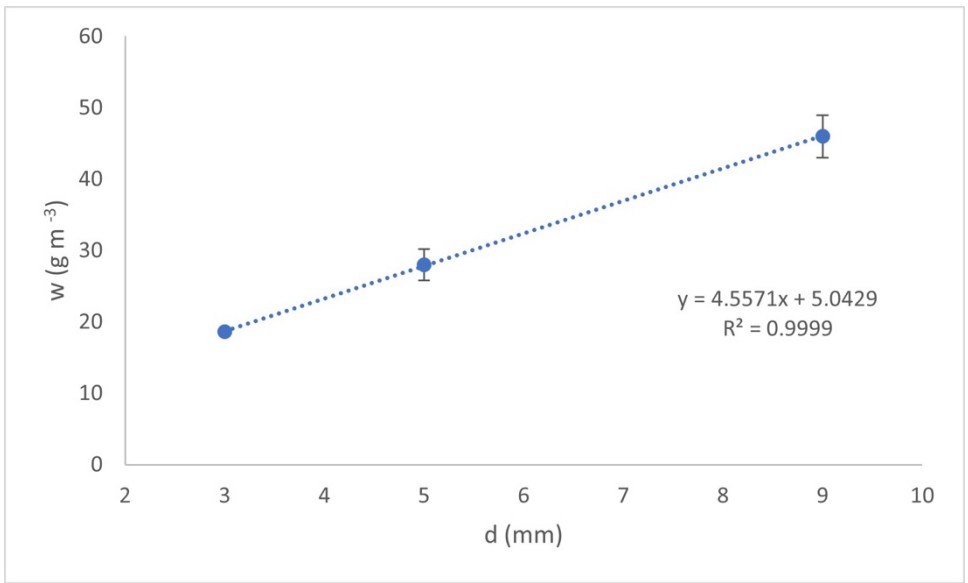

**Figure 3.** Effect of the interelectrode distance on the amount of aluminum hydroxides formed, to treat $F^-$ contaminated water.

As shown in Figure 3, the increase in formed aluminum hydroxides is almost linear with d. The interelectrode distance with the lowest amount of aluminum hydroxides formed was that corresponding to $d = 3$ mm, followed by $d = 5$ mm and finally $d = 9$ mm, with 18.5, 28, and 46 g $m^{-3}$, respectively. This is because reactions (1) and (2) are carried out instantaneously; however, the reactions (3) and (4) take more time due to their dependence on the MT. Due to the above, the influence of contaminant adsorption depends strongly on the dispersion of the flocs generated in the reactor in comparison to the amount of them. Therefore, the dispersion of $Al(OH_3)$ flocs generated in the reactor chamber is more important than the amount generated in the chamber.

Figure 4 shows the effect of the interelectrode distance on the potential drop at the terminal electrodes at the time of treatment in which an $F^- = 1.5$ mg $L^{-1}$ concentration was achieved.

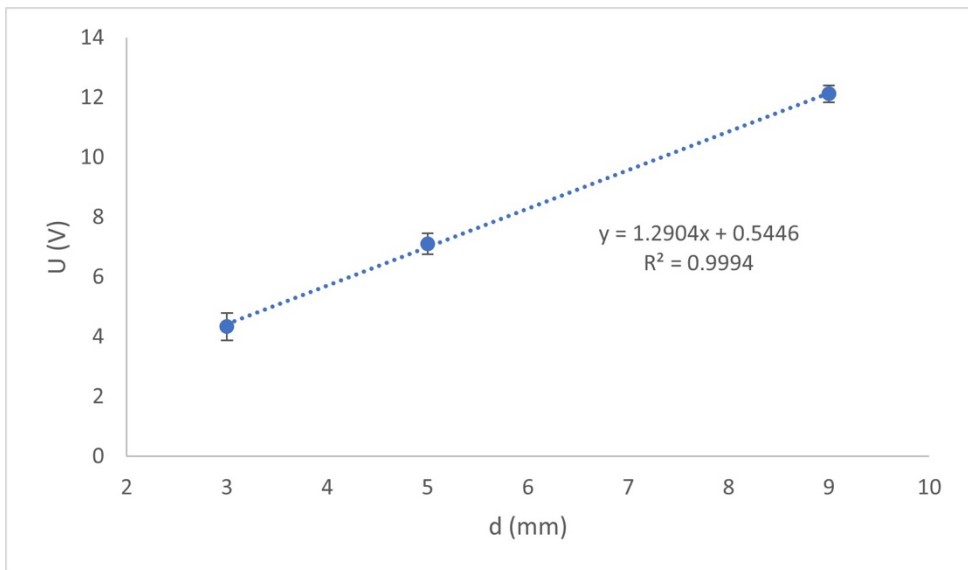

**Figure 4.** Potential drop in the terminals as a function of the interelectrode distance for the treatment of water contaminated with $F^-$.

Figure 4 shows the following: the voltage drop in the reactor increases approximately linearly with the increase in the electrode distance. This tendency can be explained in terms of Equation (6) presented above, which indicates that $U$ depends directly on the interelectrode distance.

Figure 5 shows the effect of the interelectrode distance in the energy electric power consumption (E) required to achieve an $F^-$ concentration of 1.5 mg $L^{-1}$ in the treated water.

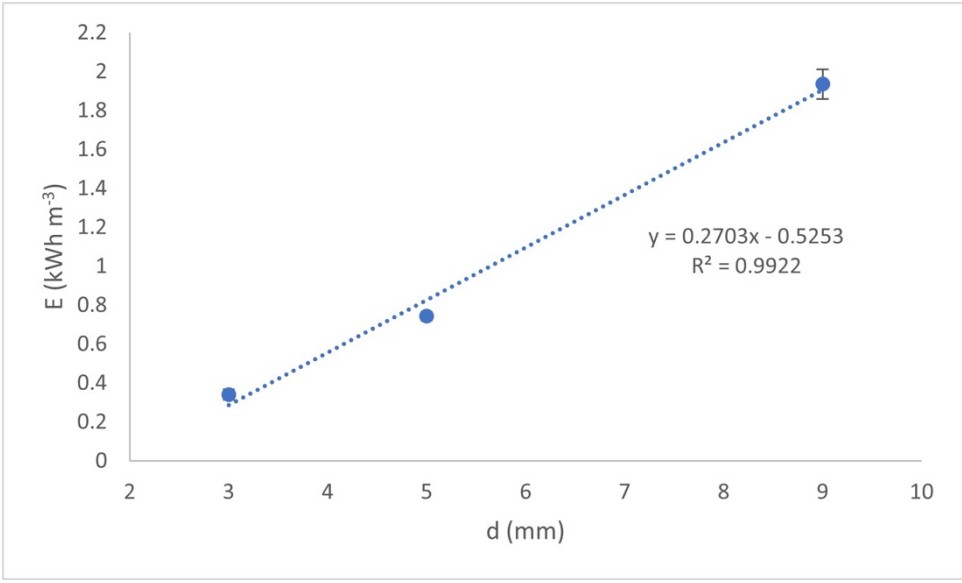

**Figure 5.** Effect of the interelectrode distance on E.

In Figure 5, the increase in the electricity consumption is not only directly proportional to the interelectrode distance but also to the treatment time, which is consistent with Equations (6) and (7) previously presented. It is also observed that this increase is approximately linear. It should be noted that this energy consumption involves that consumed by the reactor and by the air pump used to produce the movement of the liquid (3 watts per hour).

## 8. Simulation of the Developed Velocity Profiles

To visualize and numerically determine the velocity profiles achieved by the liquid inside the EC reactor at different interelectrode distances, the software FLUENT version 6.3 was employed. The software supports the numerical simulations by the Reynolds Stress Equation model (RSM), analyzing the flow and trajectory of the liquid. The simulation validates the water's velocity profiles inside of the electrocoagulation reactor for different interelectrode distances, considering a steady state. The input parameters were water velocity, density, and viscosity; walls and electrodes were considered as solid materials.

Figure 6 shows the simulation results of the axial velocity profiles developed by the water inside the reaction chamber when the interelectrode distances are: (a) 3 mm, (b) 5 mm, and (c) 9 mm.

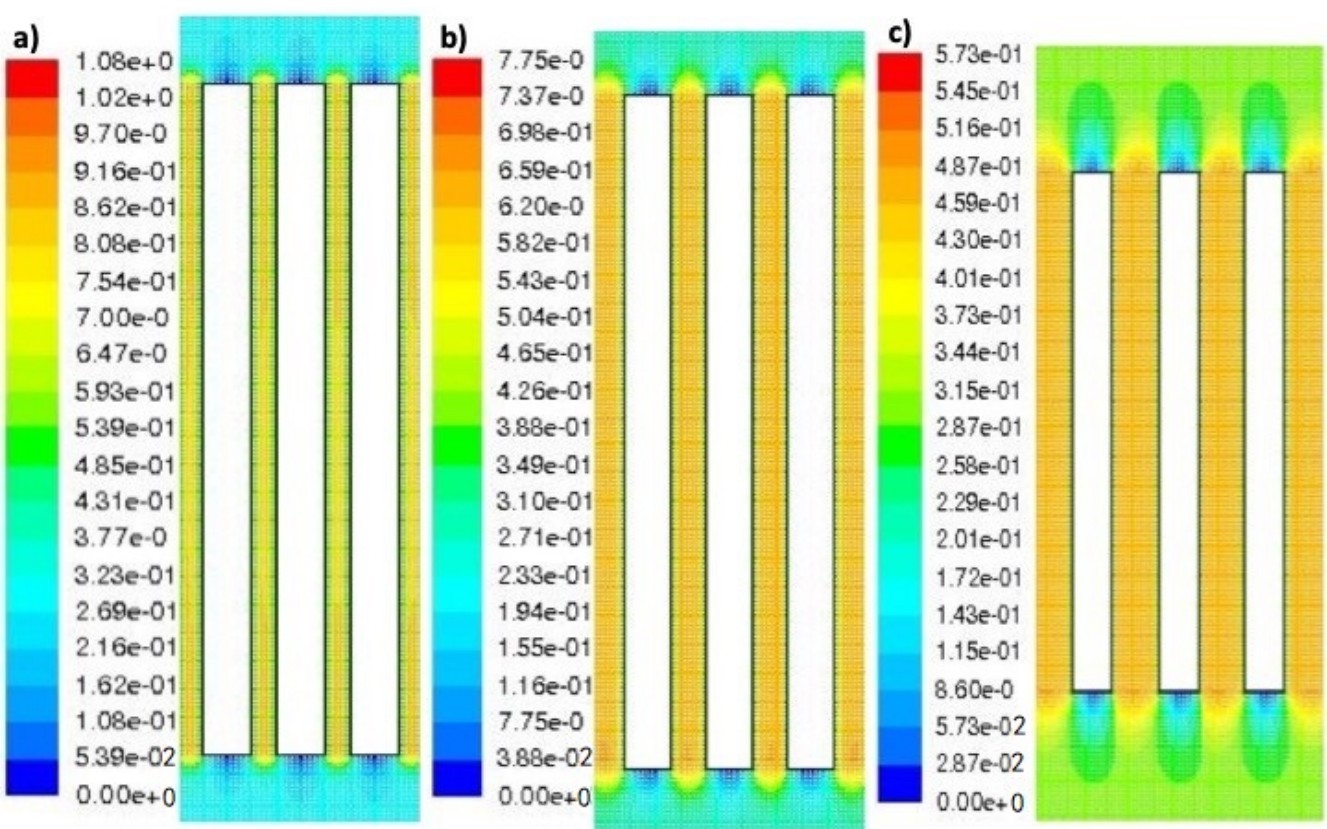

**Figure 6.** Velocity profiles developed by the water in the reaction chamber when the interelectrode distances are (**a**) 3 mm, (**b**) 5 mm, and (**c**) 9 mm.

As can be seen in Figure 6a–c, in which the white rectangles simulate the electrodes and the colors, the liquid motion. The initial and final velocity of the liquid moving through the reactor chamber is $0.3\ \text{ms}^{-1}$, the liquid velocity at the inferior and superior base of the electrodes is $0\ \text{ms}^{-1}$, and the velocity component of the liquid is faster in the interelectrode spaces. The initial velocity increases within the interelectrode spaces to approximately 0.48, 0.65, and $0.95\ \text{ms}^{-1}$ for interelectrode distances of 9, 5, and 3 mm, respectively. Therefore, it can be established that the movement of the species participating in the EC process (reaction (1)–(4)) is faster when d decreases, due to the velocity component of the liquid through the reactor chamber directly influencing the mass transfer of the process. These results show that the interelectrode distance has great importance not only in the formation velocity of flocs ($AlOH_3$) but also in the fluoride removal by them.

Figure 7 shows the maximum liquid velocity (Vmax) reached within the individual cells as a function of the interelectrode distance value.

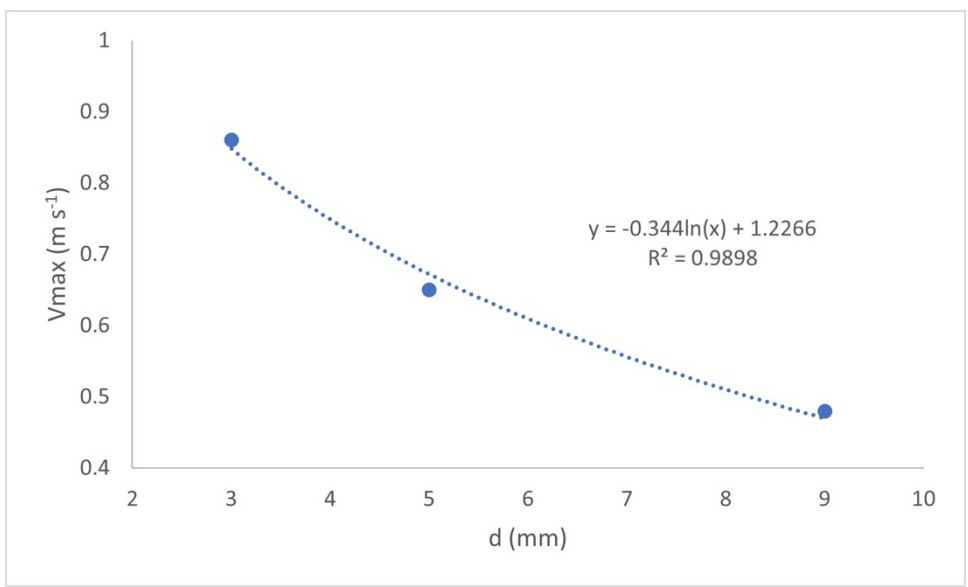

**Figure 7.** Maximum liquid velocity (Vmax) reached within the individual cells as a function of the value of the interelectrode distance.

Figure 7 shows the following, Vmax decreases as the value of d increases, from 0.86 m s$^{-1}$ for $d$ = 3 mm, to 0.48 m s$^{-1}$ for $d$ = 9 mm, which directly influences the mass transfer of the reactor, which represents the movement of the species participating in the EC process (reactions (1)–(4)). It is also observed that the values of Vmax are adjusted precisely to a logarithmic line ($R^2$ = 0.9898) with the relation, Vmax = $-0.344\ln(x) + 1.2266$.

## 9. Conclusions

The liquid velocity in the reaction chamber increases as the interelectrode distance decreases, achieving a greater mass transfer in the EC reactor. The interelectrode distance has great importance not only in the formation velocity of flocs (AlOH$_3$) but also in the fluoride removal by them. With shorter interelectrode distance, the EC reactor not only consumes less electrical energy but also fewer electrodes, and the dispersion of generated flocs in the reactor chamber is major. The dispersion of generated flocs Al(OH$_3$) in the reactor chamber is more important than the amount of flocs generated.

**Author Contributions:** Conceptualization, J.F.M.-V., J.G.-C. and N.A.R.-R.; Data curation, J.F.M.-V. and O.G.-C.; Formal analysis, J.F.M.-V., J.S.G.-V. and O.G.-C.; Funding acquisition, C.R.M.-V.; Investigation, F.C.-R. and C.R.M.-V.; Methodology, J.S.G.-V., N.A.R.-R. and O.G.-C.; Resources, J.C.O.-C. and F.C.-R.; Supervision, J.C.O.-C., F.C.-R. and J.G.-C.; Validation, J.C.O.-C., J.G.-C. and C.R.M.-V.; Visualization, J.C.O.-C. and N.A.R.-R.; Writing—original draft, J.F.M.-V. and C.R.M.-V.; Writing—review & editing, J.S.G.-V. All authors have read and agreed to the published version of the manuscript.

**Funding:** This research received no external funding.

**Data Availability Statement:** The data and methods used in the research are presented in sufficient detail in the document for other researchers to replicate the work.

**Acknowledgments:** The authors gratefully acknowledge the financial support of PRODEP and CONACYT Mexico.

**Conflicts of Interest:** The authors declare no conflict of interest.

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
