# Peer review of "Interelectrode Distance Analysis in the Water Defluoridation by Electrocoagulation Reactor"

_sustainability, doi:10.3390/su141912096_

Round 1
Reviewer 1 Report
This manuscript investigated the distance between electrodes in a reaction chamber, executing an electrocoagulation process. Various experiments were conducted with varying distance: 3, 5, 9 mm, while treating water contaminated with fluoride (F). Furthermore software fluent was used to simulate the water velocity. The authors concluded that liquid velocity increases as the electrode distance decreases. Similarly shorter electrode distance leads to EC reactor consuming less electrical energy but also fewer electrodes.
The current work is interesting. However lots of important information is missing and a major revision is advised:
1) Lines 20-21: “The software FLUENT … reactor chamber”. Possibly substitute “employee” with “employed”.
2) Line 62: The symbol “w” shown in equation (1) is not defined in lines 63-66, or elsewhere.
3) Line 64: “…and (M) is the atomic mass”. The authors should explain further of which substance the “atomic mass” refers to.
4) Line 92-94: “Which one treats a …in bipolar mode”. Maybe substitute “Which one” with “Each one”.
5) Line 117: “… and greater mass transfer ... is achieve in the…”. Set the correct tense. Change “achieve” to “achieved”.
6) Lines 130-131: “The following figure 3…linear with d”. Here the authors are introducing figure 3 for a second time. Figure 3 has already being introduced in Lines 124-126. Thusly lines 130-131 and 124-126, should be merged.
7) The authors in the current study are investigating the effects of electrode distance in an electrocoagulation process. However their conclusions indicate that shorter distance is always better. This seems to be a one sided result, that doesn’t account for the size of the reactor chamber. If the water body that needs to be decontaminated increases (maybe a few meters or more) then it becomes really hard for two electrodes with distance of 3 mm to efficiently remove the fluoride from the whole reactor chamber. Greater electrode distances would increase the range of decontamination and surely decrease the time of the whole process. The authors should include such a discussion in their manuscript that accesses more objectively the efficiency of electrode distance. However if the authors consider their results are as they should be, they should add references and extensive discussion from literature that supports their arguments.
8) The authors are using Fluent to simulate the water movement. However it seems that they provide inadequate information about their simulations. They omit a) the type of model they used (steady state or not), b) what type of equations used (they don’t have to include them in the manuscript, just name them), c) what kind of parameters obtained from the literature for their simulations and d) which parameters were used from the current study. This information is necessary for the readers to access the accuracy of the resulting simulations.
9) Lines 199-200: “The dispersion of generated flocs… flocs generated”. This is a very important statement. Dispersion of Al(OH3) is of great significance for the effective decontamination process. Discussion should be included to support this statement. This discussion is related to remark (8) stated above.
Author Response
Added an attached file with the answers to your comments.

Reviewer 2 Report
This manuscript described a electrochemical method to defluoride water and quantitative analysis of the fluid velocity with respect to various interelectrode distances. Although the idea of this manuscript is fine, the authors need to improve the quality of the content of this manuscript including experimental design and detailed discussion before it can be published. Here are my comments:
1. The authors mentioned the electrochemical method of water defluoridation, but there is no CVs (cyclic voltammetry) performed for any of the electrochemical processes. Please study the reaction at anode and cathode and the defluoridation steps in detail, including reversibility, reaction rate, etc.
2. For the experiments in figure 2-5, replicates are highly recommended to check the repeatability of the process. Please add error bar on the plots after completing the repeated experiments.
3. The simulation section needs to be greatly improved. When the reaction is taken place, there will be huge amount of H2 bubbles generated at the cathode, the simulation needs to take this into account for electrode area reduction, etc. Other parameters the authors need to consider are the change of pH during the reaction, precipitate forming.
Author Response

(The authors gave the same response as above.)

Round 2
Reviewer 1 Report
Could have being better, but I am satisfied with revision.
Author Response
Results after appropriate observations provided a detailed and more complete manuscript. Thank you very much for your kind attention.
Reviewer 2 Report
The manuscript needs major revision before it can be published and I don't see the revision I suggested has been addressed.
Author Response
Results after appropriate observations provided a detailed and more complete manuscript:
To comment one, this research is focused on the analysis of electrocoagulation and the most characteristic in this type of study are the reactions in the anode, cathode, and defluorination (which are found in lines 44-56 with their respective references), due to the above, the other studies were not considered in this research.
To comment two, all the experiments in this investigation were carried out in triplicate, and the plotted results correspond to the experiment's average. To avoid this confusion and following your recommendation, error bars were added to the graphs (Fig. 3-5). In Figure 2 the error bars are not observed, because during treatment time there is a minimal variation, and therefore the results are not identified in the graphs.
To comment three, As can be seen in reaction 2 of the research (line 49) at the cathode H2O is reduced forming OH- and H2 bubbles, the latter flow instantaneously to the surface of the liquid during the process, therefore, not reaching to minimize the area of the electrode (cathode) since, the reaction is continuously occurring. Moreover, Figure 3 discusses the formation of precipitates (aluminum hydroxides), which is the fundamental part of this research.
Thank you very much for your kind attention.